# Multicolor Tunable Electrochromic Materials Based on the Burstein–Moss Effect

**DOI:** 10.3390/nano13101580

**Published:** 2023-05-09

**Authors:** Xia Zhou, Enhui Huang, Rui Zhang, Hui Xiang, Wenying Zhong, Bo Xu

**Affiliations:** 1Key Laboratory of Biomedical Functional Materials, School of Science, China Pharmaceutical University, Nanjing 211198, China; xzhou322@163.com (X.Z.); huangenhui97@163.com (E.H.); zrui_cpu2023@163.com (R.Z.);; 2School of Mathematics and Physics, Hubei Polytechnic University, Huangshi 435003, China

**Keywords:** electrochromic (EC) materials, multicolor tunable, Burstein–Moss effect

## Abstract

Inorganic electrochromic (EC) materials, which can reversibly switch their optical properties by current or potential, are at the forefront of commercialization of displays and smart windows. However, most inorganic EC materials have challenges in achieving multicolor tunability. Here, we propose that the Burstein–Moss (BM) effect, which could widen the optical gap by carrier density, could be a potential mechanism to realize the multicolor tunable EC phenomenon. Degenerated semiconductors with suitable fundament band gaps and effective carrier masses could be potential candidates for multicolor tunable EC materials based on the BM effect. We select bulk Y_2_CF_2_ as an example to illustrate multicolor tunability based on the BM effect. In addition to multicolor tunability, the BM effect also could endow EC devices with the ability to selectively modulate the absorption for near infrared and visible light, but with a simpler device structure. Thus, we believe that this mechanism could be applied to design novel EC smart windows with unprecedented functions.

## 1. Introduction

Electrochromic (EC) materials, which can dynamically modulate colors (optical properties) in response to electrical signals, have attracted intense scientific and technological interest due to their promising applications in the field of displays and smart windows [1,2]. Many inorganic and organic materials are developed as EC materials, such as transition metal oxides [3,4,5,6,7,8] and organic molecules/metal complexes/conjugated polymers (CPs) [9,10]. Among them, inorganic EC materials have attracted the most attention, because they offer better durability, thermal stability, and chemical stability than organic EC materials, especially in the applications for outdoor smart windows [11]. However, for most inorganic EC devices, it is still a challenge to achieve multicolor tunability [12,13]. Recently, structurally color-enhanced EC devices have been suggested as an effective mechanism to overcome this challenge. Based on this mechanism, various strategies [14,15,16,17,18,19], including photonic crystals, plasmon resonances, and interference effects, have been pursued in designing multicolor tunable devices. However, these structural colors are extremely sensitive to the feature size of nanostructures, requiring precise lithography or high-vacuum deposition techniques, which inherently restrict their cost reduction. Therefore, it is crucial to develop a more economic and feasible strategy or mechanism to broaden the color adjustment based on inorganic EC materials.

Organic EC materials show significant advantages in color tuning over inorganic materials. Organic EC materials usually have two or more reversible oxidation/reduced states, which have different light absorptions, especially in certain wavelengths [9,20,21]. Thus, these oxidation/reduced states exhibit different colors, which are mainly determined by the energy level systems for light absorption, as illustrated in Figure 1a,b. In principle, organic EC materials could show multiple colors upon external stimulations, if the reversible oxidation/reduced states are enough. However, until now, there have been few organic EC materials with several oxidation/reduced states [3]. By extending the mechanism of multicolor tunability in organic EC materials, if one material contains a series of tunable energy levels in the visible light range, it could be a potential multicolor tunable EC candidate.

For degenerated semiconductors, their optical properties, including the colors, are dependent on their optical band gaps. In many transparent conducting oxides (TCOs), the Burstein–Moss (BM) effect has been proved as an effective method to modify the optical band gaps by carrier concentration [22,23]. As illustrated in Figure 1c, the BM effect can be expressed as: *E_opt_* = *E_g_ + E_BM_*, where *E_opt_* is the optical band gap, *E_g_* is the fundamental band gap between conduction band (CB) and valence band (VB), and *E_BM_* is BM shift due to carrier doping. The BM shift can broaden the optical band gaps by increasing the carrier density. The tunable optical band gap in the visible range could change the color if the BM shift is large enough. In some TCOs, the BM shift can up to 0.8 eV [24]. However, TCOs do not show the color change because of their large fundamental band gaps. If a semiconductor possesses a suitable fundamental band gap and a large BM shift, that could broadly modulate its optical band gap in the visible light range upon carrier density. Accordingly, the optical absorption would be adjusted, especially in certain wavelengths, which would result the color tunability, as illustrated in Figure 1d. Thus, it is quite possible to realize multicolor tunable EC phenomenon based on the BM effect. However, to the best of our knowledge, this strategy has never been used in multicolor tunable EC devices.

Here, we propose that the BM effect would be a potential mechanism for multicolor tunable electrochromism, and also suggest that degenerated semiconductors with suitable electronic properties could be potential candidates for multicolor tunable EC materials. Y_2_CF_2_ is selected as an example to reveal the multicolor tunable EC phenomenon based on the BM effect. With the electron doping, the color of Y_2_CF_2_ will change to yellow and green from orange. More interestingly, the BM effect is also found to be useful for the designing of dual-band EC smart windows, because significant plasmonic absorption for near-infrared (NIR) can be simultaneously induced with the color change.

## 2. Calculation Methods

The electronic properties of bulk Y_2_CF_2_ are calculated by the Vienna ab initio Simulation Package (VASP) [25,26] based on density functional theory (DFT). The electronic structures are described by using the generalized gradient approximation (GGA) based the Perdew–Burke–Ernzerhof (GGA-PBE) exchange–correlation functional [27] with the projector augmented wave (PAW) [28]. The cutoff energy used for the plane–wave basis set as 500 eV for bulk Y_2_CF_2_. The gamma-centered 18 × 18 × 10 k-meshes were adopted to sample the first Brillouin zone. The structures were relaxed until the atomic forces were less than 0.01 eVÅ^−1^ per atom and the total energies converged to 10^–5^ eV. Hybrid functional (HSE06) is applied to compute the electronic structures of bulk Y_2_CF_2_, which generally yield more reliable band energies [29].

The optical properties of Y_2_CF_2_ are calculated according to the dielectric functions. The imaginary part of the dielectric function contributed by interband transition is calculated by [30]:(1)$$\varepsilon_{2}^{inter}\left(\omega \right)=\frac{4\pi^{2}e^{2}}{\Omega }\underset{q\to 0}{\lim}\frac{1}{q^{2}}\sum_{c,v,k}2w_{k}\delta (E_{ck}-E_{vk}-\omega )\times {\left|\langle u_{ck+q}|u_{vk}\rangle \right|}^{2}$$
where e is the electron charge, Ω is the primitive cell volume, and w_k_ is the weight of the k-points. The indices c and v refer to the CB and VB states, respectively. The parameter E_jk_ is the single-electron energy state of band j at wave vector k; u_jk_ is the periodic part of the Bloch wave function corresponding to the eigenvalue E_ik_ (i = c, v); and δ is the delta function, which depends upon the method used for calculation. The real part of the dielectric function contributed by interband transition is obtained from the Kramers–Kronig relation.

The optical absorption for NIR light is due to the plasmonic absorption, which could be determined by dielectric functions from intraband transitions, ε^intra^(ω). The contributions from intraband transitions, ε^intra^(ω), are obtained by the Drude expression for a given plasma frequency ω_p_ and damping constant (γ) [31]:(2)$$\varepsilon^{intra}=\varepsilon_{\infty }-\frac{\omega_{p}^{2}}{\omega^{2}+i\gamma \omega }$$
while ω_p_ can be calculated from the electronic band structure as follows:(3)$$\omega_{p\alpha \alpha }^{2}=-\frac{8\pi e^{2}}{V}\sum_{n,k}^{}v_{nk_{\alpha }}^{2}\frac{\partial f_{nk}}{\partial \epsilon_{nk}}.$$

## 3. Results and Discussion

To be a multicolor tunable EC material based on BM effect, E_opt_ should be varied in the visible range as broadly as possible. If E_opt_ can vary from 1.2 eV to 3.2 eV, the color of the material would change from red, to orange, yellow, green, and, at last, transparent. Thus, the optimized E_g_ should be about 1.2~2.0 eV, and the BM shift should be as large as possible to make E_opt_ cover the visible range. Theoretically, the BM shift is dependent on the reduced effective mass (μ) and the carrier density (n): ℏ22μ(3π2n)23 [24]. As we can see, to obtain a large BM shift at a certain carrier density, a small reduced effective mass is needed. However, to display the color (light absorption) corresponding to the band gap, a considerable reduced effective mass is needed. These two contradictory requirements can be balanced by the semiconductor with a flat valence band (VB) and a well-dispersed conduction band (CB). The flat VB can endow the considerable optical absorption around the optical band gap, while the well-dispersed CB can ensure enough BM shift. Moreover, we should notice that E_opt_ is the interband transition from VB to CB as shown in Figure 1c. After electron doping, the interband transition could also occur between CB and CB + 1. To keep the BM shift as the main reason for color change, the energy between CB + 1 and CB around the CB minimum should be large enough. Thus, the essential electronic properties for a suitable multicolor tunable EC material can be summarized as: a suitable E_g_ (1.2~2.0 eV), a flat valence band but a small electron effective mass, and a large energy difference between CB + 1 and CB. Until now, there has been no report regarding the multicolor tunable EC materials based on the BM effect. We believe it is due to the rarity the materials which can satisfy these four requirements.

Being aware of possibility of the BM effect in realizing the multicolor tunable EC phenomenon, we screened potential candidates from both previous reports and material databases based on these requirements. We found that Y_2_CF_2_, one of MXenes, could be a potential candidate for a multicolor tunable EC material using the BM effect. Recently, bulk Y_2_CF_2_ was experimentally reported as a semiconductor photocatalyst with a band gap of about 1.9 eV in experiments [32,33]. According to the literature, bulk Y_2_CF_2_ shows a dispersed CB but a flat VB, which indicates that a large BM shift can be obtained in bulk Y_2_CF_2_. Here, we perform theoretical calculations on the optical properties and color changes of Y_2_CF_2_ upon electron doping. The optical absorption, especially the absorption edge, is strongly related to the band gap, while PBE functional usually underestimates the band gaps of semiconductors. Thus, we study the electronic and optical properties of Y_2_CF_2_ upon electron doping with HSE06 functionals [31].

The optimized lattice parameters of Y_2_CF_2_ (a = 3.656 Å and c = 6.291 Å) are close to the previous theoretical and experimental values reported [32,33]. As shown in Figure 2a, Y_2_CF_2_ exhibits a typical structure of MXenes. We firstly calculate the orbital projected band structure of bulk Y_2_CF_2_ in Figure 2b. As we can see, the CB is mainly composed of d orbitals of Y atoms, while the VB is dominated by p orbitals of C atoms. Without doping, bulk Y_2_CF_2_ exhibits an indirect semiconductor character: the CB minimum (CBM) is at the M point, while the VB maximum (VBM) is located at the Γ point. However, the VB at the M point is only 0.1 eV lower than the VBM at the Γ point. Thus, the optical absorption around the band gap is still dominated by the direct interband transition at the M point. The electronic gap at the M point is about 1.97 eV, which is consistent with the experimental results (1.9 eV) [24]. The VB at the M point is rather flat, and the effective electron mass is about 0.6m_0_ (m_0_ is the mass of free charge.). Meanwhile, the energy difference between CB and CB + 1 around the M point is up to 3.1 eV. Thus, Y_2_CF_2_ could be a potential multicolor tunable EC material based on the BM effect, because it fits the requirements we suggested very well.

Next, we examine the BM effect in Y_2_CF_2_ by electron doping. In principle, electron doping will gradually move up the Fermi level to cross CB, which could result the BM effect in Y_2_CF_2_. To reveal this, we also calculated the band structures of Y_2_CF_2_ with different electron doping concentrations, displayed in Figure 3a–d. Due to the flat VB at the *M* point, the energy between the Fermi level and the CB minimum can be approximated to the BM shift. Apparently, the BM shift increases with the electron doping, as shown in Figure 3e. The BM shift can reach 0.5 eV at the doping level of 0.2 electrons per unitcell (e/unitcell). As we can see, the relation between the BM shift and *n* is approximate to the theory of *n*^2/3^ at low concentrations. However, it starts to diverge from the electron concentration of 0.15 e/unitcell. The difference can reach 0.24 eV at an electron concentration of 0.2 e/unitcell. This is mainly because we use the same effective electron mass to fit the BM shift. However, the effective mass *μ* is also dependent on the carrier density, especially at high carrier density. Moreover, many-body interactions, electron–electron, and electron–impurity scattering, often shrink the fundamental band gaps, which would also partially compensate for the BM shift [24]. The many-body interactions can be enhanced at high carrier density. However, even being enhanced, the effect of many-body interactions is still much less than the BM shift in many TCOs, suggesting that the carrier doping could widen the optical band gap of Y_2_CF_2_. Meanwhile, the energy difference between CB and CB + 1 remains larger than that from VB to CB until the electron doping reaches 0.2 e/unitcell. The optical band gap is dominated by the BM shift and fundamental band gap. Due to the BM shift, the optical band gap of Y_2_CF_2_ can increase from 1.97 eV to 2.52 eV upon the electron doping, which can effectively modulate the optical absorption in the visible range, and which also may result in a color change.

By the calculated optical absorption in Figure 4a, the absorption edge for undoped Y_2_CF_2_ appears around 615 nm. We find that doping the electrons blueshifts the absorption edge. It is blueshifted to about 450 nm at the electron concentration of 0.2 e/unitcell. Such a large shift of the absorption edge, about 165 nm, would result significant color change. Typically, EC devices can be categorized into transmissive-type and reflective-type by structure. Here, we calculate color based on the transmittance for 300 nm thick Y_2_CF_2_ with different electron concentrations. The transmittance (T(λ)) of Y_2_CF_2_ is determined by: T(λ) = 1 − α(λ) − R(λ), where α(λ) and R(λ) are the absorbance and reflectance of Y_2_CF_2_, respectively. Based on T(λ), we calculate the color coordinates defined by the International Commission of Illumination (CIE 1931 XYZ) [34], which provides a mathematical foundation for a quantitative description of colors. The XYZ tristimulus values for an object illuminated by a light source with a spectral power distribution S(λ) are then given by:(4)$$\begin{array}{c} X=\int S\left(\lambda \right)T(\lambda )\bar{x}(\lambda )d\lambda ;\\ Y=\int S\left(\lambda \right)T(\lambda )\bar{y}(\lambda )d\lambda ;\\ Z=\int S\left(\lambda \right)T(\lambda )\bar{z}(\lambda )d\lambda ; \end{array}$$
where $\bar{x}(\lambda )$, $\bar{y}(\lambda )$, and $\bar{z}(\lambda )$ are three color-matching functions for red, green, and blue lights, respectively. The color coordinates in CIE XYZ color space are defined by: $x=\frac{X}{X+Y+Z}$ and $y=\frac{Y}{X+Y+Z}$. As we can see in Figure 4b, for undoped Y_2_CF_2_, the transmittance is dominated by a red component, and part of a green component, and thus appears orange. As the electron density increases to 0.05 e/unitcell, it becomes yellow due to the increase in the green component, and further changes to yellow-green at the electron density of 0.2 e/unitcell as some of the blue component is added. As we can see, the carrier concentration can effectively tune the color of Y_2_CF_2_ by modulating the optical band gap, which endows the possibility for multicolor tunable EC devices based on the BM effect.

To observe the multicolor tunable EC phenomenon based on the BM effect, it is vital to fabricate EC devices which can modulate the carrier concentration using an electrical potential. For traditional inorganic EC devices [3], their optical responses are usually controlled by the intercalation and deintercalation processes of small ions (H^+^, Li^+^, etc.) under an electrical potential. In some sense, intercalating small ions into traditional inorganic EC materials would be regarded as tuning the carrier density in EC devices. It would be very convenient if we could use traditional inorganic EC devices to realize the multicolor tunable EC phenomenon. However, intercalating ions usually results in volume expansion, which may narrow the fundamental band gaps. We intercalate Li^+^ ions into Y_2_CF_2_ with different concentrations, Li_x_Y_2_CF_2_ (x = 1/4, 1). It was found that intercalating Li^+^ ions would make volume expansion. For Li_1/4_Y_2_CF_2_, the optimized lattice parameters are a =3.5556 Å and c= 7.712 Å. As a result, the volume is expanded about 15% compared to undoped Y_2_CF_2_. The volume becomes much bigger with the concentration increase. As a consequence, the fundamental band gap is reduced to 1.5 eV for Li_1/4_Y_2_CF_2_ (Figure 5a). As x =1, the fundamental band gap decreases, and the direct optical transition has been changed to the M point between CB and CB + 1, as denoted by the green arrow. As we can see, intercalating Li^+^ ions could decrease the fundamental band gap, which tends to compensate the BM shift. Thus, traditional inorganic EC devices based on intercalation and deintercalation of small ions may not suitable for the EC devices using the BM effect.

But luckily, field effect transistors (FETs) are the most commonly used electronic devices to modulate the carrier density. To change the color, the required electron density in Y_2_CF_2_ is 0.2 e/unitcell, about 2.8 × 10^21^ cm^−3^. Such a high carrier density is hard to realize in solid-gated FETs, but it can be obtained in some ionic liquid gated transistors [35,36]. In addition, Yi et al. [37] have reported ionic liquid-gated EC devices based on La_0_._2_Sr_0_._8_MnO_3_/SrIrO_3_ supperlattices. Thus, ionic liquid gated transistors can be used as the prototype devices for multicolor tunable electrochromism based on the BM effect, which allows to modulate the color using external gate voltages. Traditional inorganic EC devices usually have three-core-layer structures consisting of an electrochromic film, an ion conductor, and an ion storage film. However, EC devices based on BM effect only need two core layers: a semiconductor layer and an ionic liquid-gating layer. The simpler device structure is favorable to the design of EC devices with novel EC functions. Compared to traditional inorganic EC devices, the EC behaviors under electrical potentials are different. The optical transmission in the visible range decreases as the ions are intercalated for traditional inorganic EC devices. However, for the EC devices based on the BM effect, they have opposite EC phenomenon: the optical transmission increases with the carrier concentration. This unique EC behavior would endow smart windows with unprecedented EC functions.

Furthermore, we notice that the plasmonic effect is an intrinsic effect of carrier doping in the semiconductors [38], which would usually induce significant plasmonic absorption for NIR. The plasma frequency ω_p_ can be given by ω_p_^2^ = ne^2^/ε_0_m^*^, in which n is the carrier density and m^*^ is the effective mass of carrier. As we can see, the screened plasma frequency ($\omega_{p}^{*}=\omega_{p}/\sqrt{\varepsilon_{\infty }}$) would increase along with the electron concentration, which is identified by the calculation for Y_2_CF_2_ (Figure 6a). The screen plasma frequency can be increased to 0.92 eV at the electron doping concentration of 0.2 e/unitcell, which could significantly enhance the optical absorption for NIR (Figure 6b). For Y_2_CF_2_, the phonons occur below 1000 cm^−1^ (10 μm). We only consider the dielectric function to 5 μm. We do not need to consider the phonon contributions for dielectric function here. Thus, EC devices based on the BM effect can simultaneously modulate the transmittance in both the visible and NIR ranges by tuning the carrier density. Increasing the electron density in Y_2_CF_2_ would blueshift the absorption edges in both the visible and NIR ranges. It suggests that the EC devices based on the BM effect could be potential dual-band EC devices, which have emerged as a promising technology to reduce building energy consumption [39,40]. In recent reported dual-band EC devices, NIR, containing about 50% of the total solar radiation, could be blocked by sacrificing part of the transparency. However, transparency is an important performance for building windows [41,42]. EC devices based on the BM effect can block the NIR and increase the transparency at the same time. This behavior would provides a great opportunity to develop energy-efficient dual-band EC applications.

However, Y_2_CF_2_ is not the best candidate for multicolor tunable EC material based on the BM effect. For Y_2_CF_2_, the color can only vary from orange to yellow-green in EC devices. In addition, the required electron density to tune the color is quite high, up to 2.8 × 10^21^ cm^−3^. Both of these factors limit the practical applications of Y_2_CF_2_ in multicolor tunable EC devices. The limited color adjustment is mainly attributed to the large fundamental band gap of Y_2_CF_2_, about 2.0 eV. To enrich the color tunability, the best fundamental band gap should be around 1.5 eV. The high required electron density is due to the large effective mass of electron, about 0.7m_0_. A small effective mass of electrons is preferable to obtain a large BM shift at a certain carrier density. For Sb doped SnO_2_ [43], a BM shift of about 0.5 eV can be observed at the carrier density of 4 × 10^20^ cm^−3^, about one order less than that of Y_2_CF_2_, due to the small reduced mass around 0.2m_0_. However, smaller is not better, because we need a high optical absorption corresponding to the optical band gap. Thus a semiconductor with a band gap around 1.5 eV and a moderate reduced mass around 0.2~0.4m_0_ is preferable for multicolor tunable EC devices. Thus, discovering suitable candidates plays an important role in applying the BM effect in multicolor tunable EC devices. We believe that better candidates could be discovered by high-throughput screening based on the material databases, which have been successfully applied to discover functional materials.

## 4. Conclusions

In summary, we suggest that the BM effect induced by carrier doping could be a mechanism for the multicolor tunable EC phenomenon, and Y_2_CF_2_ is selected as the candidate to demonstrate this property. In addition to the multicolor tunability, the BM-effect-based EC devices also display potential applications in energy-efficient dual-band EC devices with simpler device structures. Our findings reveal a hitherto-unexplored strategy to develop multicolor tunable EC smart windows, which shows potential in the application of multifunctional EC devices.

## Figures and Tables

**Figure 1 nanomaterials-13-01580-f001:**
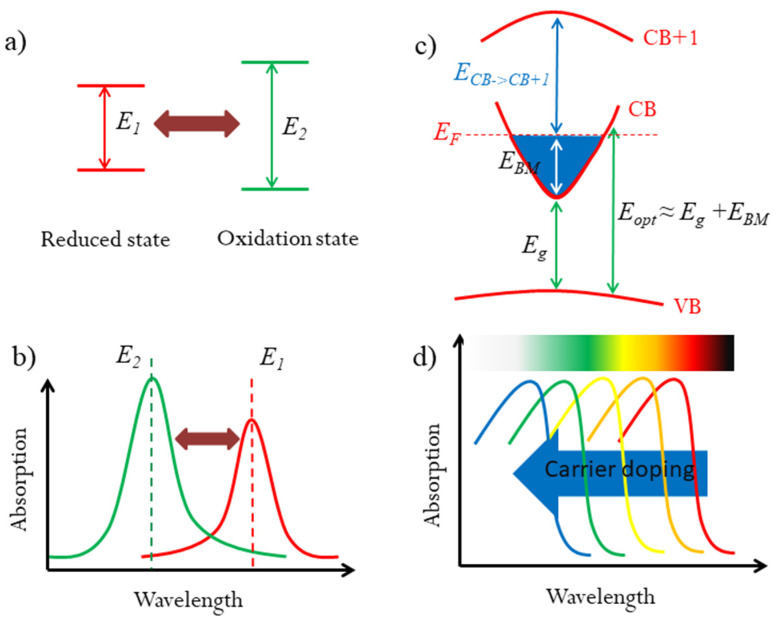
Strategy for multicolor tunable EC materials. (**a**) The mechanism for organic color tunable EC materials: different transition energy for reduced and oxidation states, resulting in different optical absorption wavelengths. (**b**,**c**) Diagram of the optical widening effect of the Moss–Burstein shift by carrier doping in degenerated semiconductors. (**d**) The optical absorption would be blueshifted in the visible range for suitable *E_g_* and *E_BM_*, which could result in the color change.

**Figure 2 nanomaterials-13-01580-f002:**
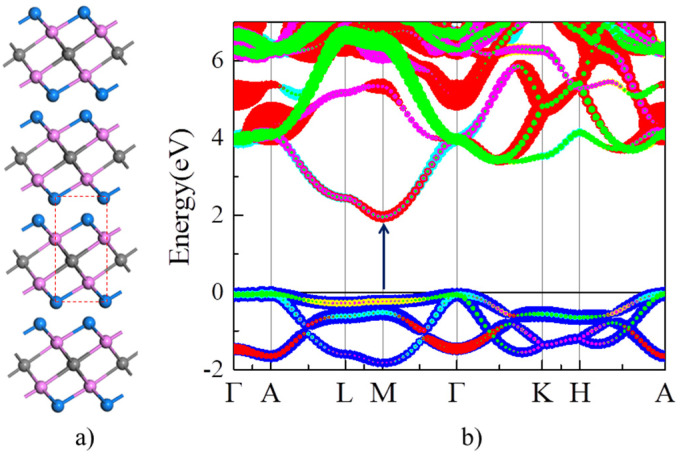
Structure and electronic structure of Y_2_CF_2_. (**a**) Structure of Y_2_CF_2_. Gray, purple, and blue balls represent C, Y, and F atoms, respectively. The dashed line denotes the unit cell of Y_2_CF_2_. (**b**) The orbital projected band structure of Y_2_CF_2_. The blue lines represent *p* orbital of carbon, while other color lines represent *d* orbital of Y.

**Figure 3 nanomaterials-13-01580-f003:**
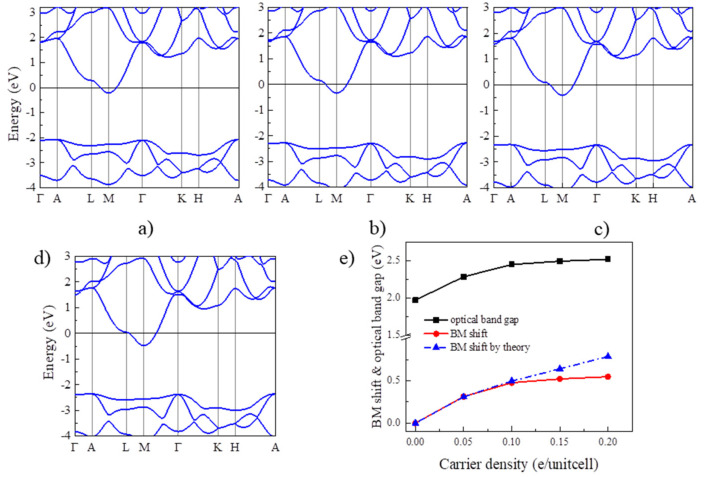
BM shift by electron doping for Y_2_CF_2_. (**a**–**d**) are the band structures of electron doped Y_2_CF_2_ with concentration of 0.05, 0.1, 0.15, and 0.2e/unitcell, respectively. (**e**) The calculated BM shift (red line) and the optical band gap (black line) under electron doping. The blue line is the theoretical BM shift according to the law of *n*^2/3^.

**Figure 4 nanomaterials-13-01580-f004:**
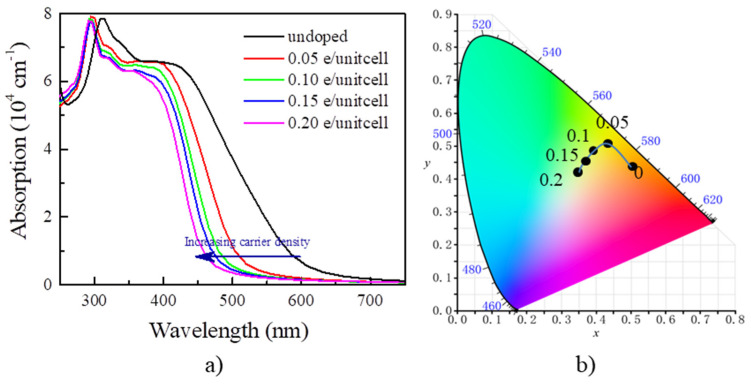
The optical absorption and color for Y_2_CF_2_ under electron doping. (**a**) The blueshift of optical absorption by electron doping; (**b**) the color coordinates of the transmittance of 300 nm thick Y_2_CF_2_ in CIE XYZ color space.

**Figure 5 nanomaterials-13-01580-f005:**
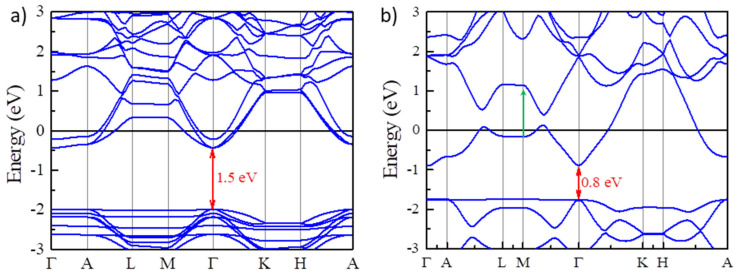
Narrowing the fundamental band gaps by intercalating Li^+^ ion into Y_2_CF_2_. (**a**,**b**) the electronic band structures of Y_2_CF_2_ with different Li^+^ concentrations Li_x_Y_2_CF_2_ (x = 1/4, 1). The red arrows indicate the fundamental band gaps for different Li^+^ concentrations.

**Figure 6 nanomaterials-13-01580-f006:**
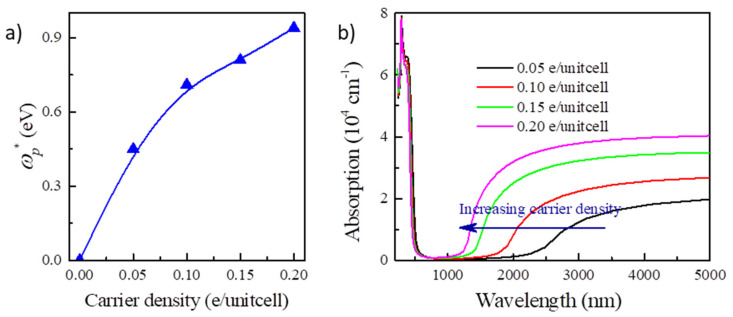
NIR light absorption in electron doped Y_2_CF_2_. (**a**) The dependence of screen plasma frequency $\omega_{p}^{*}$ of Y_2_CF_2_ on carrier density. (**b**) NIR light absorption of Y_2_CF_2_ upon different electron concentrations.

## Data Availability

The data that support the findings of this study are available from the corresponding author upon reasonable request.

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
