# Peer review of "Multicolor Tunable Electrochromic Materials Based on the Burstein–Moss Effect"

_nanomaterials, 2023, doi:10.3390/nano13101580_

Round 1

Reviewer 1 Report

The manuscript titled Multicolor tunable electrochromic materials based on Burstein- 2 Moss effect by Xia Zhou et al, is a nice study that may be well accepted by a broad community. However, it requires revision prior to taking any decision. Please, see the comments below.

Titled: It is ok.

Abstract: Missing information about stability and reproducibility.

Introduction: When referring to the present state of the art, worth to mention the role that the structure and composition may have on the performances achieved See Wojcik, PJ et al in Microstructure control of dual-phase inkjet-printed a-WO3/TiO2/WOX films for high-performance electrochromic applications, 2012 | JOURNAL OF MATERIALS CHEMISTRY, 22 (26) , pp.13268-13278 and in Tailoring nanoscale properties of tungsten oxide for inkjet printer electrochromic devices, 2015 | NANOSCALE, 7 (5) , pp.1696-1708).

Moreover, worth to refer the electrochromic transistor (Grey, P et al in Solid State Electrochemical WO3 Transistors with High Current Modulation, Sep 2016 | ADVANCED ELECTRONIC MATERIALS, 2 (9)).

Experimental section:

How many samples were processed? How reproducible ad reliable the process and the films are? What is the error associated when evaluating films processed in the same batch but in different spatial locations? What are the errors associated with performances from batch to batch? What are the environmental conditions in which the devices were tested? Did you notice ageing effects? Did you notice any type of persistent photoconductive effect, namely connected to UV? Any thermalization effects noticed.

Results and discussion:

Well done but requires some more information on current modulation.

Conclusions: Overall good. Missing quantitative key performance indicators of the work performed.

Figures: Are OK.

Table: Missing a table with the key performance indicators of the study performed

References: require updated.

Author Response

Dear Sir/Madam:

Thank you very much for your comments on our manuscript entitled as “Multicolor tunable electrochromic materials based on Burstein-Moss effect” (Paper # nanomaterials-2318684). We appreciate you for carefully reading our manuscript. Your comments are very constructive and valuable for our further research. Based on your comments and our new results, we have modified our manuscript. The modifications have been marked in red.

So, we would like to resubmit our manuscript to Nanomaterials. We hope that you can reconsider our revised manuscript for publication. We also expect that our revised manuscript can convince you and be acceptable to Nanomaterials.

Best wishes,

The authors

Reviewer 2 Report

The Manuscript entitled, ‘Multicolor tunable electrochromic materials based on Burstein- Moss effect’ by Zhou et al. describes a strategy of a multi color electrochromic device based on the principle of Burstein-Moss shift of the optical band gap. The material studied in this work is Y2CF2. The authors have used first-principles calculations to model the electronic and optical band structure of the material. However, there are several shortcomings and it is difficult to positively evaluate the manuscript in the present format. Please consider the questions below.

1)      Since the authors are modelling the electronic structure, they should provide the lattice parameters of the unit cell that they calculate and correlated them to the experimental data of the material. This includes the volume of the unit cell also.

2)      If there are severe deviations between the theoretical and experimental lattices then please use the Hubbard model to refine these values.

3)      Figure caption of figure 3 is incomplete. I can not understand the differences between (a), (b), (c) and (d). Please revise this so that I can give my feedback.

4)      How was the BM shift obtained in theory? Please give differences between the blue and red lines in 3e. How was the optical band gap obtained?

5)      Please provide lattice and volume changes on intercalation with H+ and Li+ obtained from calculations. Also comment on the band gap that now appears to be direct at x=1 at the M-point. This will affect greatly the absorption properties.

6)      The paper should be more coherently and clearly written. There is a mention of electron doping initially, how was this done? just by adding electrons or some metal doping is considered? Then the material is doped with protons to reduce the gap. But this not clearly explained and it is very confusing.

7)      Please read and correct typos.

8)      Correlate the results obtained to experimental works and compare if bandgap changes have occurred with electron or proton doping in those works.

9)      Since the LSPR energies are quite far from the interband transitions, the authors have to consider that there are several phonon modes in this region. Have the authors considered this in their calculations? The dielectric function in this region would become quite complex.

Author Response

(The authors gave the same response as above.)

Round 2

Reviewer 1 Report

The paper was properly revised and deserves publication now

Reviewer 2 Report

The authors have made the changes as suggested. However, comments 6 and 9 are not discussed in the paper and should be presented somewhere in the text.
